# Antimicrobial Efficacy and Spectrum of Phosphorous-Fluorine Co-Doped TiO_2_ Nanoparticles on the Foodborne Pathogenic Bacteria *Campylobacter jejuni*, *Salmonella* Typhimurium, Enterohaemorrhagic *E. coli*, *Yersinia enterocolitica*, *Shewanella putrefaciens*, *Listeria monocytogenes* and *Staphylococcus aureus*

**DOI:** 10.3390/foods10081786

**Published:** 2021-07-31

**Authors:** György Schneider, Bettina Schweitzer, Anita Steinbach, Botond Zsombor Pertics, Alysia Cox, László Kőrösi

**Affiliations:** 1Department of Medical Microbiology and Immunology, Medical School, University of Pécs, Szigeti Street 12, H-7624 Pécs, Hungary; scbtabt.pte@pte.hu (B.S.); anitani88@gmail.com (A.S.); pertics.botond@gmail.com (B.Z.P.); 2Department of Biotechnology, Nanophagetherapy Center, Enviroinvest Corporation, Kertváros Street 2, H-7632 Pécs, Hungary; alcox@tcd.ie; 3Research Institute for Viticulture and Oenology, University of Pécs, Pázmány Péter Street 4, H-7634 Pécs, Hungary; korosi.laszlo@pte.hu

**Keywords:** antibacterial, foodborne, nanoparticles, photocatalysis, co-doping, TiO_2_

## Abstract

Contamination of meats and meat products with foodborne pathogenic bacteria raises serious safety issues in the food industry. The antibacterial activities of phosphorous-fluorine co-doped TiO_2_ nanoparticles (PF-TiO_2_) were investigated against seven foodborne pathogenic bacteria: *Campylobacter jejuni*, *Salmonella* Typhimurium, Enterohaemorrhagic *E. coli*, *Yersinia enterocolitica*, *Shewanella putrefaciens*, *Listeria monocytogenes* and *Staphylococcus aureus*. PF-TiO_2_ NPs were synthesized hydrothermally at 250 °C for 1, 3, 6 or 12 h, and then tested at three different concentrations (500 μg/mL, 100 μg/mL, 20 μg/mL) for the inactivation of foodborne bacteria under UVA irradiation, daylight exposure or dark conditions. The antibacterial efficacies were compared after 30 min of exposure to light. Distinct differences in the antibacterial activities of the PF-TiO_2_ NPs, and the susceptibilities of tested foodborne pathogenic bacterium species were found. PF-TiO_2_/3 h and PF-TiO_2_/6 h showed the highest antibacterial activity by decreasing the living bacterial cell number from ~10^6^ by ~5 log (*L. monocytogenes*), ~4 log (EHEC), ~3 log (*Y. enterolcolitca*, *S. putrefaciens*) and ~2.5 log (*S. aureus*), along with complete eradication of *C. jejuni* and *S.* Typhimurium. Efficacy of PF-TiO_2_/1 h and PF-TiO_2_/12 h NPs was lower, typically causing a ~2–4 log decrease in colony forming units depending on the tested bacterium while the effect of PF-TiO_2_/0 h was comparable to P25 TiO_2_, a commercial TiO_2_ with high photocatalytic activity. Our results show that PF-co-doping of TiO_2_ NPs enhanced the antibacterial action against foodborne pathogenic bacteria and are potential candidates for use in the food industry as active surface components, potentially contributing to the production of meats that are safe for consumption.

## 1. Introduction

Foodborne pathogenic bacteria can seriously influence the safety and quality of meats. They not only can cause diseases and death, but also represent an economic burden [1]. Some species are normal microbiota members present on livestock or the skin of humans and animals, while others are ubiquitous in the surrounding environment of the processed animal.

*Campylobacter jejuni*, *Salmonella* Typhimurium, Enterohaemorrhagic *E. coli* (EHEC), *Yersinia enterocolitica*, *Shewanella putrefaciens*, *Listeria monocytogenes* and *Staphylococcus aureus* are among the most impactful foodborne bacterial pathogens. *C. jejuni* is one of the most common causative agents of bacterial foodborne illness, mostly associated with the consumption of undercooked poultry meat and milk [2]. The non-typhoid Salmonella Typhimurium is considered to be the second most important bacterial foodborne infection of the western world [3]. It is widely known that egg and chicken consumption are the main risk factors for salmonellosis, but seafoods, other meats and vegetables play also a role in the persistent occurrence of outbreaks [4]. EHEC is one causative agent of haemorrhagic colitis (HC) and haemolytic uraemic syndrome (HUS), two life-threatening conditions in humans caused by this pathogen. Contaminated and improperly cooked beef is a major source of EHEC infection in humans [5]. *Y. enterocolitica* is less known, but still the fourth bacterial foodborne zoonosis, most commonly reported in the European Union (EU) [6]. Pigs are considered the main reservoirs, with pork products acting as major sources for human infections [7]. *S. putrefaciens* is a typical spoilage bacterium of seafood products [8] but was also described as the causative agent of soft tissue infections and bacteraemia in humans [9,10]. The ubiquitous *L. monocytogenes* is the causative agent of listeriosis, commonly associated with hard cheeses and bovine meat [11]. The incidence of human listeriosis cases is low. However, based on the severity of disease and the likelihood of mortality upon infection, the risk associated with this pathogen is high [11]. Food poisoning is frequently associated with the presence of enterotoxin-producing bacterium species. Foods contaminated with strains of *Staphylococcus aureus* are one of the most common causes of food poisoning [12]. Raw meat and meat products are ideal food matrices favouring their growth. Their presence in food can cause food poisoning, septicaemia, toxic shock syndrome and fatal endocarditis [13].

Intestinal microbiota, skin of animals and humans and the environment itself are habitats of potential foodborne pathogens [1]. Artificial surfaces such as equipment used for slaughtering, and packaging materials, can be permanent or transient habitats on which bacteria can adhere, survive and proliferate, thereby acting as sources of bacterial foodborne pathogen contamination [1,14]. To solve this detrimental problem, endowing surfaces with antibacterial properties has been recognized an effective way to preventing bacterial adhesion and subsequent biofilm formation [15].

Due to their practical significance, increased attention has recently been paid to nanoparticles (NPs) that show antibacterial activity. The antibacterial activity of various metal oxide NPs originate from their photocatalytic properties [16]. Titanium dioxide (TiO_2_) is one of the most studied photocatalysts [17,18,19]. Reactive oxygen species (ROS) such as superoxide (O_2_^•−^) and hydroxyl radicals (OH^•^) can be generated on photocatalytically active TiO_2_ NPs by photons with appropriate energy, most commonly with ultraviolet (UV) light [20,21]. The produced ROS can effectively degrade harmful compounds and inactivate microorganisms [22].

Furthermore, TiO_2_ is a commonly used and safe food ingredient (E171) that has been approved for use by the United States Food and Drug Administration (FDA) (2015). TiO_2_ is also useful when applied as a photocatalyst in packaging materials [23,24].

The photoinduced antimicrobial activity of various semiconductor metal oxides has been studied intensively. Due to its low cost and high photocatalytic activity, TiO_2_ and anatase polymorph, is considered to be an ideal candidate for antimicrobial technologies [25]. The nanosized TiO_2_ has several advantageous physicochemical characteristics. The large surface area favours an increased rate of photocatalysis [26]. To enhance their photocatalytic and antibacterial activity, the surface of TiO_2_ can be doped or modified with compounds including noble metals and metal oxides such as silver, copper oxide, vanadium, chromium, lanthanum and many others [27,28,29,30,31,32] via different synthetic procedures such as solvothermal-, sol-gel- and microwave methods [33,34,35]. The photocatalytic activity of TiO_2_ NPs can be further improved by co-doping, a novel technique using two different dopants together during synthesis [36,37]. Even though metals are effective candidates, their application raises toxicity issues in several cases, precluding their use in food applications. Therefore, doping or co-doping of TiO_2_ with non-metal elements, such as nitrogen [38,39], are frequently preferred. However, reports on non-metal doped or co-doped TiO_2_ NPs are still scarce.

One such study investigated the antimicrobial efficacies of undoped, doped or co-doped TiO_2_ NPs on different meats such as lamb and fish [40,41]. Their antibacterial efficacies were revealed by targeting only a few important foodborne pathogen bacteria: *Listeria monocytogenes*, *Salmonella* Typhimurium, *Campylobacter jejuni* [42,43]. Until now, no systematic study has been carried out examining the survival of the most important foodborne pathogenic bacteria in the presence of co-doped TiO_2_ NPs.

In a recent study we produced and thoroughly characterized the physical and chemical properties of PF-co-doped TiO_2_ NPs, where both P and F were used as dopants, and revealed their enhanced antibacterial activities against carbapenem resistant *Klebsiella penumoniae* isolates [44]. In order to investigate the possible applicability of PF-co-doped TiO_2_ in the food industry, here we have compared the antibacterial efficacy of PF-co-doped TiO_2_ NPs on seven foodborne pathogenic bacterium isolates: *C. jejuni*, *S.* Typhimurium, Enterohaemorrhagic *Escherichia coli*, *Y. enterocolitica*, *S. putrefaciens*, *L. monocytogenes* and *S. aureus*. With these tests we examined both the pure photoinduced antimicrobial efficacies and how the duration of the hydrothermal treatments influenced the antibacterial features of the produced NPs on multiple foodborne pathogenic bacteria.

## 2. Materials and Methods

### 2.1. Synthesis and Characterization of PF-Co-Doped TiO_2_ NPs

Synthesis of PF-co-doped TiO_2_ NPs (PF-TiO_2_) was performed as described previously [44]. Briefly, 46 g of TiCl_4_ was suspended in 100 mL of 2-propanol and sonicated for 5 min. The light yellow solution was diluted with 200 mL of highly purified deionized water. These Ti precursors were hydrolysed by adding 500 mL of 1.5 M NaOH solution dropwise under vigorous stirring. The resulting white precipitate was centrifuged and washed three times with 100 mL of deionized water. After that, 0.38 mol/L HPF_6_ solution (7.3 mL) was added to the dispersion while stirring. Aliquots (50 mL) of the dispersion were treated hydrothermally at 250 °C for 1, 3, 6 or 12 h. Samples were denoted PF-TiO_2_/0 h, PF-TiO_2_/1 h, PF-TiO_2_/3 h, PF-TiO_2_/6 h and PF-TiO_2_/12 h, respectively. The dispersions were then centrifuged and washed with deionized water followed by 2-propanol. The obtained sediments were dried at 50 °C in an oven for 12 h.

### 2.2. Bacterium Strains and Growth Conditions

For the antibacterial tests, the following foodborne pathogens were used: *Campylobacter jejuni* (NCTC 11168), *Salmonella enterica* subsp. enterica serovar Typhimurium (ATCC 14028) (*S.* Typhimurium), Enterohaemorrhagic *Escherichia coli* EDL 933 (ATCC 43895), *Yersinia enterocolitica* subsp. *enterocolitica* O8 (ATCC 23715), *Shewanella putrefaciens* (ATCC BAA-453), *Listeria monocytogenes* (ATCC 35152) and *Staphylococcus aureus* (ATCC 23235). All bacteria were routinely grown in Mueller Hinton broth or on Mueller Hinton Agar (Oxoid, USA), except for *C. jejuni*. For this species, the charcoal cefoperazone deoxycholate agar (CCDA) was used as a growth vehicle. All bacteria were grown at 37 °C, except *C. jejuni* (42 °C) and *S. putrefaciens* (30 °C). Plates were incubated for 24 h under aerobic conditions, except *C. jejuni* which was incubated for 48 h under microaerophilic conditions.

### 2.3. Antibacterial Tests

In order to compare the susceptibility of seven foodborne pathogens against different PF-co-doped TiO_2_ NPs, antibacterial tests were performed. A starter culture was made on the day of the test from an overnight culture of the relevant bacterium grown in 20 mL of MH medium as described in Section 2.2. At mid logarithmic phase (OD_600_ = 0.4–0.6), the culture of the relevant bacterium was centrifuged 10,000× *g* for 2 min, and then washed twice with 0.9 *w*/*v*% NaCl solution. At the last washing step, the suspension was centrifuged, and the optical density was set to OD_600_ = 1 (1 × 10^8^ Colony Forming Units (CFU)/mL) in 0.9 *w*/*v*% NaCl solution. Tests were carried out in a flat bottom non-adhesive 24-well tissue culture plates containing 990 μL of dispersion of PF-co-doped TiO_2_ NPs in 0.9 *w*/*v*% NaCl and 10 μL of bacterium suspension. The final concentration of the bacterium suspension was ~10^6^ CFU/mL. Three different concentrations (500 μg/mL, 100 μg/mL, 20 μg/mL) of the PF-co-doped TiO_2_ NPs were compared for their antibacterial efficacies.

After thorough mixing by pipetting with a 1 mL tip, PF-co-doped TiO_2_ and bacterial suspensions were incubated in a closed dark box for 15 min. Then, the photocatalytic reactions were carried out using a 15-W UV-A lamp (F15W/T8/BL368 fluorescent lamp, Sylvania, Wilmington, MA, USA). For bacterial enumeration, 10 μL of sample aliquots were taken after 15 min dark incubation and 30 min after UV exposure. Experiments were also performed under daylight conditions by using a 15 W lamp (Viva-Lite T8/W15/5500K), or under dark conditions for 30 min.

In every test, concentration-matched bacterium suspensions in 0.9 *w*/*v*% NaCl solution without any PF-TiO_2_ NPs served as bacterium controls, while AEROXIDE TiO_2_ P 25 (Evonik, Essen, Germany) acted as a control.

After incubation, the numbers of viable cells were determined by dropping and running off 10 μL of suitably diluted aliquots onto MH agar plates and then counting the colonies after 24 h incubation at the appropriate temperature (30, 37 or 42 °C) and growth condition (aerobic or microaerophilic), depending on the bacterium species under investigation. Each test was performed in triplicate and repeated on a separate day.

### 2.4. Statistical Analysis

Statistical analysis was performed using Microsoft Excel 2016 MSO software (Microsoft Corp., Redmond, WA, USA). Students *t*-test was applied to all pairwise comparisons. The level of significance was *p* < 0.05. Data were expressed as mean ± standard deviation.

## 3. Results and Discussion

The antibacterial activity of PF-co-doped TiO_2_ NPs on *Campylobacter jejuni*, *Salmonella* Typhimurium, Enterohaemorrhagic *E. coli*, *Yersinia enterocolitica*, *Shewanella putrefaciens*, *Listeria monocytogenes* and *Staphylococcus aureus*, which are among the most important bacterial foodborne pathogens, were studied [2,3,5,7,10,12,14]. The applied PF-TiO_2_ NPs were characterized recently [44]. Therefore, this study has strictly focused on the comparative antibacterial analysis of the NPs on the afore-mentioned foodborne pathogenic bacteria. The main structural and morphological features of PF-TiO_2_ NPs are summarized in Figure 1. X-ray powder diffraction measurements revealed that the crystalline phase of all PF-TiO_2_ samples was solely the anatase itself (JCPDS card No. 78-2486). The prolonged hydrothermal treatment resulted in higher intensities of d_101_ diffraction peaks while the peak broadening decreased gradually over time (Figure 1a). These changes in the peak shape indicate that hydrothermal treatment enhanced the crystallinity and increased the crystallite size of PF-TiO_2_ samples. The average crystallite size was calculated from the full width at half maximum (FWHM) of d101 peak by using Scherrer equation. Without treatment, ~6 nm anatase crystallites were formed while treatments for 1–12 h resulted in 10–24 nm crystallites (Figure 1b). In addition to producing larger crystallites, treatment also increased the anatase content, thus the proportion of amorphous phase reduced (Figure 1b). It should be noted that even without hydrothermal treatment, high crystallinity was observed (76 wt%). The high crystallinity of the samples is crucial for the effective photocatalytic generation of ROS [45]. Crystal defects determine the photoelectric properties of semiconductors, and they can significantly affect the charge carrier recombination [46]. Figure 1c,d shows representative TEM images of PF-TiO_2_/0 h and PF-TiO_2_/12 h samples. For the untreated sample, tiny particles are barely distinguishable (Figure 1c). After 12 h hydrothermal treatment, mainly spherical but also anisotropically shaped nanocrystals are visible (Figure 1d). In accordance with the XRD measurements, TEM images also showed that the particle size increased sharply during the hydrothermal treatment, however, the mixed morphology was retained. Consequently, PF-TiO_2_ nanoparticles with both spherical and polyhedral (faceted) shapes have been observed for all the treated samples [44]. TEM images suggest that polyhedral nanoparticles possess truncated tetragonal bipyramidal geometry, which is typical for the anatase nanocrystallites.

Results of antibacterial tests show that none of the bacterium species were sensitive to the PF-TiO_2_ NPs at any concentrations under dark conditions, indicating that their antibacterial effects stem from photocatalytic reactions (Figure 2). This is in agreement with previous studies [44] where TiO_2_ NPs did not exhibit antibacterial effects under dark conditions [38,39,42,43]. Without light excitation, TiO_2_ NPs were not toxic for bacteria as CFU values indicated (Figure 2). Under dark conditions, TiO_2_ NPs can adsorb on the cell wall of bacteria. In order to ensure adsorption–desorption equilibrium, PF-co-doped TiO_2_ NPs had to be thoroughly mixed with bacterial suspensions under dark conditions and incubated for 15 min. The duration of this phase depends on the applied NP and typically ranges from 10–40 min [38]. However, light with appropriate energy can induce charge separation in the TiO_2_ NPs and subsequently redox reactions can take place on their surface. Recently, we studied the formation of the main ROS in the PF-TiO_2_ dispersions by means of electron paramagnetic resonance (EPR) spectroscopy. During the photocatalytic reactions, the presence of hydroxyl (OH^•^) and superoxide radicals (O_2_^•−^), as well as singlet oxygen (^1^O_2_) was confirmed [44]. Among them, the most reactive and damaging OH^•^ radical was the predominant form. Similar to our recent study [44], high bacterial cell number (10^6^ CFU/mL) was applied in order to reveal antibacterial efficacy of TiO_2_ NPs, although such a high bacterial CFU is usually not present on the surface of meats.

Figure 2 displays that the antibacterial activity of PF-TiO_2_ NPs depends on the (i) duration of hydrothermal treatment, (ii) the concentration of NPs and (iii) the susceptibility of the foodborne pathogenic bacterium species.

Even though TiO_2_ are considered safe in food applications by the FDA, no comparative study has been reported on the antibacterial potential of TiO_2_ NPs on various foodborne pathogen bacteria. Even though some data are available for *L. monocytogenes*, *S.* Typhimurium, *C. jejuni*, *S. aureus* [42,43,47,48,49], there is no available data for EHEC, *Y. enterocolytica* and *S. putrefaciens*. These studies focused on the photocatalytic-based antibacterial activity of standard nano-TiO_2_ (P25) [42,47,48], or its combination with Ag, an antibacterial metal [43]. Even though the enhanced antibacterial efficacies of TiO_2_ NPs doped with other metals and metal oxides, including Cr, Cu, CoO and ZnO, were demonstrated under laboratory conditions [19,50,51,52,53], their application raises some toxicity issues. Several metals are not only toxic for the bacterial cell, but also for humans and eukaryotic organisms [53,54,55,56,57,58].

In the present study, the most sensitive bacterium species to PF-co-doped TiO_2_ NP was the Gram-negative *C. jejuni*, that was unable to survive at any of the tested NP concentrations (500 μg/mL, 100 μg/mL, 20 μg/mL) during UVA exposure (Figure 2a, Appendix A). Only this bacterium showed a significant decrease of CFU during daylight conditions at all concentrations (Appendix A). Its living cell number dropped from 2 × 10^6^, to 3 × 10^5^, 7 × 10^4^, 4 × 10^4^, 6 × 10^4^ and 1 × 10^5^ with the application of PF-TiO_2_/0 h, PF-TiO_2_/1 h, PF-TiO_2_/3 h, PF-TiO_2_/6 h, PF-TiO_2_/12 h NPs, respectively (Figure 2a). Furthermore, UVA alone (in the absence of NPs) caused a >1 order of magnitude decrease in the CFU of this species (Figure 2a). Our results are only partially comparable with earlier findings where Ag, ZnO and CuO NPs were investigated on *C. jejuni* and found that the MBC values of these NPs were 50 μg/mL, 100 μg/mL and 5000 μg/mL, respectively [59], as authors applied a 100-fold less concentrated (10^4^ CFU/mL) bacterium suspension. Furthermore, the exposure times were 16 h instead of the 30 min applied in our study [59].

A plausible explanation for the extreme susceptibility of *C. jejuni* is its lack of the classical oxidative stress response regulatory elements SoxRS and OxyR [60]. Our current results indicate that the defence mechanism of *C. jejuni* is unable to cope with the ROS generated by PF-co-doped TiO_2_ NPs. To best of our knowledge no previous study has investigated the survival of this bacterium species under direct UVA exposure. However, our data are partially comparable with a previous study [61] where a 5 log CFU/mL reduction was observed after the application of 405 nm light for 30 min in a suspension-based model. Even though we did not see a 5 log CFU/mL reduction in our experiments, but we demonstrated that PF-co-doped TiO_2_ NPs drastically enhance the antibacterial effects of UVA treatments.

*S.* Typhimurium was the second most sensitive bacterium species, showing significant bacterial cell number loss when incubated with PF-TiO_2_ NPs (Figure 2b, Appendix A). The most effective photocatalytic NPs were PF-TiO_2_/3 h and TiO_2_/6 h if applied together with UVA, as they completely eradicated *S*. Typhimurium (Figure 2b). Coincubation and UVA treatment of this bacterium species with PF-TiO_2_/1 h or PF-TiO_2_/12 h resulted in a drastic 5 log CFU reduction from 2 × 10^6^, to 2 × 10^1^ and 1 × 10^1^, respectively. PF-TiO_2_/0 h was more effective than the TiO_2_ P25 (Figure 2b). The antibacterial effect of the NPs decreased with decreasing NP concentration. Concerning the comparison of the sensitivities of *C. jejuni* with *S.* Typhimurium to NPs, our results are in agreement with the results of the available published studies which showed that the sensitivity of *C. jejuni* was 8- to 16-fold lower than that of *S.* Typhimurium [59,62].

Until now, very few results are available based on the antibacterial effect of UVA (315–400 nm) as research mostly focuses on the antibacterial efficacy of UVC [49,63,64] with a wavelength in the germicide range (200–280 nm). A recent study, however, has investigated the UVA-induced antibacterial effect of TiO_2_ [42]. The experimental set-up was almost identical to that used by us allowing direct comparison of results. Long et al. [42] found that 500 μg/mL TiO_2_ P25 decreased the living *S.* Typhimurium cell count by nearly 1.5 log in 30 min, in line with our results. However, our results went beyond this as we showed that antibacterial activity of a TiO_2_ NP can be further enhanced through PF-co-doping, from among PF-TiO_2_/3 h and PF-TiO_2_/6 h formulations were the most effectives. Antibacterial activity of these co-doped, NPs were almost comparable to the bactericide activity of CuO and was better than that of ZnO demonstrated by a recent study [59]. It should be however noted that in this study the applied bacterial number was less (10^4^ CFU/mL) and the exposure times were longer (16 h) in contrast to our experimental setup where the bacterial concentration was 10^6^ CFU/mL while the exposition times were 30 min. Silver NPs, one of the most widely used antimicrobial nanomaterials [65], had the highest anti-microbial efficacy according to this study.

*C. jejuni* was the only tested species where the survival was also affected by the applied UV light alone. A 3.21 and 3.63 log CFU/g decrease in *S.* Typhimurium and EHEC, respectively, was achieved on the surface of cheese if UVA was applied in combination with citric acid [66,67]. This evidence generally supports our results, although the experiments were performed on solid surfaces, while ours employed a mixed liquid system.

The rate and pattern of antibacterial effects and bacterial sensitivity to PF-TiO_2_ NPs between the pork-associated *Yersinia enterocolitica* and the fish-associated *Shewanella putrefaciens* were very similar. Among the tested Gram-negative bacteria these two species were the least sensitive. For *Y. enterocolitica*, the initial CFU decreased from 2 × 10^6^ to 1 × 10^4^, 2 × 10^3^, 1.5 × 10^3^, 1.5 × 10^3^, 2 × 10^4^ if 500 μg/mL of PF-TiO_2_/0 h, PF-TiO_2_/1 h, PF-TiO_2_/3 h, PF-TiO_2_/6 h, PF-TiO_2_/12 h were used, respectively. Similarly, for *Shewanella putrefaciens*, the initial CFU decreased from 2 × 10^6^, to 2.5 × 10^4^, 1.5 × 10^4^, 2 × 10^3^, 1.2 × 10^3^ and 4 × 10^3^ if the PF-TiO_2_/0 h, PF-TiO_2_/1 h, PF-TiO_2_/3 h, PF-TiO_2_/6 h and PF-TiO_2_/12 h were applied together with UVA irradiation for 30 min (Figure 2d,e). None of these species was sensitive under daylight and dark conditions. Concerning these two bacterium species, very few data is available that focuses on their elimination with either UV or plasma treatments [64,68].

In contrast, being one of the most serious foodborne pathogens under EU surveillance [6], several *Listeria monocytogenes* eradication practices have been investigated based on germicidal UVC irradiation [66,69] or on UVA-combined photocatalysis [42,66]. Our results clearly demonstrated the drastic antibacterial activity of the PF-TiO_2_ NPs on *L. monocytogenes* (Figure 2f, Appendix A) and this species was the most sensitive bacterium species followed *C. jejuni* and *S.* Typhimurium. Concerning the photocatalytic activity of TiO_2_ P25, our results match with the previous results of Long et al. [42] as after 30 min incubation a 2 log magnitude decrease in the bacterial count of *L. monocytogenes* was detected, however this decrease in the living bacterial cell numbers could be further enhanced if the PF-TiO_2_ NPs, especially PF-TiO_2_/3 h and PF-TiO_2_/6 h (Figure 2f), were applied in the photocatalytic reaction. In a previous study, doping of TiO_2_ with CoO resulted in a <100 log decrease of the *L. monocytogenes* cells in 24 h when a 4 mg/mL NP suspension was applied at a 10^6^ CFU/mL final bacterium concentration [50]. Even though this antimicrobial effect of CoO was revealed both under dark and light conditions, it was undetectable after 6 h exposure.

Sensitivity of the other tested Gram-positive bacterium species *S. aureus* (Figure 2g, Appendix A) to the PF-TiO_2_ NPs was most comparable to the patterns of *Y. enterocolitica* and *S. putrefaciens*, with almost all NPs able to induce a 1–3 log CFU decrease. Due to its great importance in human health, possible elimination of *S. aureus* has previously been investigated by other authors. By using UVA irradiated TiO_2_-based photocatalysis in submerged coated systems, a ~1 log reduction in CFU was observed after 4 h [48], a result that was in line with our control (TiO_2_ P25) results. Another study showed that silver doped TiO_2_ nanotubes (Ag/TNTs) had superior antibacterial features with agar diffusion, compared to TiO_2_ (P25) and TNT alone [70]. Unfortunately, we could not find any data in the study if silver alone, as control, was tested or not. It would have been necessary as the antibacterial feature of silver is well known. A more relevant study was presented by Jalvo et al. [47], where a 3 log reduction in the living cell number in a *S. aureus* biofilm was obtained by using a TiO_2_-activated glass surface and 2 h irradiation treatments with 290–400 nm light. This study demonstrated that photocatalytically active surfaces have great potential in food industrial applications to hinder the contamination of meats with foodborne pathogenic bacteria. Moreover, doping with non-metal elements such as P and F, is a promising method to increase the antimicrobial efficacy of TiO_2_ NPs, as we have demonstrated here. Highly crystalline PF-TiO_2_ NPs with an anatase crystal phase proved to be good candidates for prevention of foodborne illness as they have superior antimicrobial effects.

## 4. Conclusions

All seven tested bacterium species are typical foodborne pathogenic bacteria that can enter the food chain during meat processing and proliferate during transportation and storage. One option to eliminate these pathogens, or at least decrease their number, is the use of antibacterial surfaces both on the tools used for processing and the food packaging itself. In this study, we have demonstrated that PF-co-doped TiO_2_ NPs have a wide antibacterial spectrum and display enhanced light inducible antibacterial activity. Therefore, they are potential candidates for application in the food industry as active surface components.

## Figures and Tables

**Figure 1 foods-10-01786-f001:**
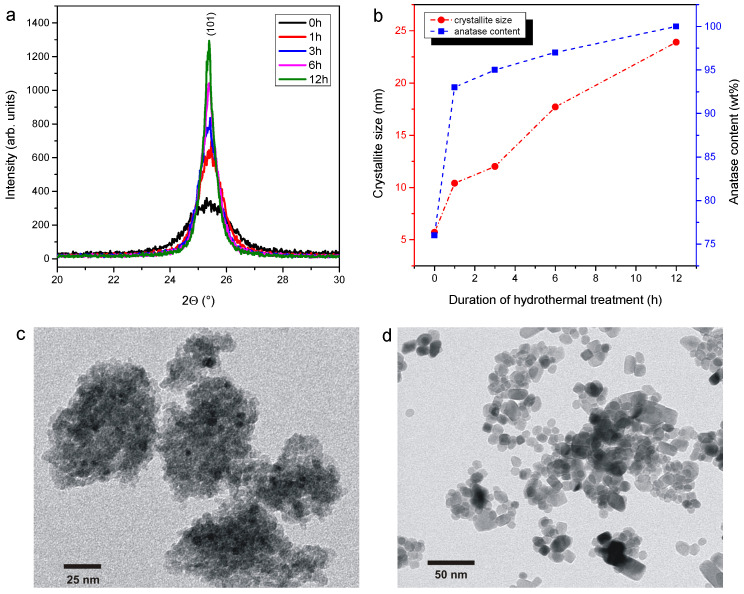
Structural and morphological properties of PF-co-doped TiO_2_ nanoparticles: X-ray diffractograms (**a**); crystallite size and anatase content of PF-TiO_2_ nanoparticles (**b**); representative TEM images of PF-TiO_2_/0 h and PF-TiO_2_/12 h samples (**c**,**d**).

**Figure 2 foods-10-01786-f002:**
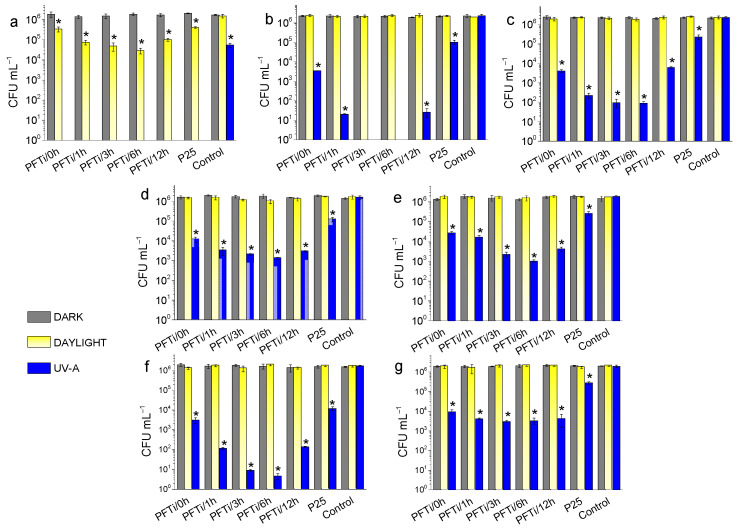
Antibacterial activities of the hydrothermally prepared PF-co-doped TiO_2_ nanoparticles on the foodborne pathogenic bacteria *Campylobacter jejuni* (**a**), *Salmonella Typhimurium* (**b**), Enterohaemorrhagic *E. coli* (**c**), *Yersinia enterocolitica* (**d**), *Shewanella putrefaciens* (**e**), *Listeria monocytogenes* (**f**) and *Staphylococcus aureus* (**g**). Experiments were performed under dark conditions or daylight or UVA exposures for 30 min. Starting CFUs were ~10^6^ CFU mL^−1^. The concentration of PF-TiO_2_ NPs was 500 μg/mL. Lack of the blue column demonstrated the complete elimination of the treated bacteria. * Significant difference at *p* < 0.05.

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
