# Peer review of "Antimicrobial Efficacy and Spectrum of Phosphorous-Fluorine Co-Doped TiO2 Nanoparticles on the Foodborne Pathogenic Bacteria Campylobacter jejuni, Salmonella Typhimurium, Enterohaemorrhagic E. coli, Yersinia enterocolitica, Shewanella putrefaciens, Listeria monocytogenes and Staphylococcus aureus"

_foods, 2021, doi:10.3390/foods10081786_

Round 1
Reviewer 1 Report
In this study, the authors addressed the antimicrobial potential of Phosporous-Fluorine co-doped anatase TiO2 nanoparticles towards important foodborne pathogens. Although the inhibitory ability of these particles have been previously described against carbapenem-resistant Klebsiella pneumoniae by the same research team, the work here described represents a novel approach for their use. However, the manuscript has some issues that should be addressed:
- In the title, the nanoparticles designation must be in full and not in abbreviation.
- Salmonella nomenclature is not uniform throughout the text. The The Kauffman and White classification used in lines 29, 47, 100, 135, 175, 253, 256, 266 and 270 should also be followed in the totle and in lines 19 and 109.
- Lines 39-42 - Not all foodborne bacteria are of pathogenic nature; in fact. most microbiota members present on the livestock and in the surrounding environment of the food processing plants are commensals. This entire paragraph must be corrected.
- Change the designation "flora" to "microbiota" (lines 41 and 67).
- Lines 104-114 - The letter size used in this paragraph is different .
- Line 138 - "and" should not be in italics.
- Lines 146-157 - The authors need to justify the selection of the concentrations of both the bacterial suspensions and the nanoparticles solutions used in this study, especially since it is not expected that foods present such a high concentration of pathogenic bacteria, nor that just one bacterial species is present.
- Besides performing replicates (line 171), were the assays repeated in independent says?
- Line 178 - Although the nanoparticles have been characterized recently by the same research team, no cytotoxicity studies were performed. The authors have addressed this issue in lines 228-234, but the fact that they have chosen non-metals is not a guarantee of safety. This must be mentioned in the paper.
- The results and discussion section must be improved - the section is mainly used for results presentation, and could be further discussed considering other active surface components (considering authors statement in conclusions, lines 321-324) and not just only other TiO2 NPs.
Author Response
Response to the comments and suggestions of Reviewer #1
Manuscript Number: Foods-1258146
Title:
“Antimicrobial efficacy and spectrum of Phosphorous-Fluorine co-doped TiO2 nanoparticles on the foodborne pathogenic bacteria Campylobacter jejuni, Salmonella Typhimurium, Enterohaemorrhagic E. coli, Yersinia enterocolitica, Shewanella putrefaciens, Listeria monocytogenes and Staphylococcus aureus”
First of all, thank you for the positive feedback concerning our manuscript submitted to Foods.
You can find our responses to your comments and suggestions below, while changes are highlighted (yellow) in the text:
- In the title, the nanoparticles designation must be in full and not in abbreviation.
Answer:
We have corrected this.
- Salmonella nomenclature is not uniform throughout the text. The The Kauffman and White classification used in lines 29, 47, 100, 135, 175, 253, 256, 266 and 270 should also be followed in the title and in lines 19 and 109.
Answer:
Yes it is absolutely true. In order to make it uniform in the text, we have corrected the name of the serovar to S. Typhimurium. We used its absolute complete name (Salmonella enterica subsp. enterica serovar Typhimurium) only in section 2.2 “bacterium strains and growth conditions”.
- Lines 39-42 - Not all foodborne bacteria are of pathogenic nature; in fact. most microbiota members present on the livestock and in the surrounding environment of the food processing plants are commensals. This entire paragraph must be corrected.
Answer:
Yes, we agree with your comment. With this sentence we only wanted to focus on foodborne pathogenic bacteria. With a slight modification we have clarified this.
See: L. 39.
- Change the designation "flora" to "microbiota" (lines 41 and 67).
Answer:
Corrected.
Please see: L. 40 and L. 65.
- Lines 104-114 - The letter size used in this paragraph is different .
Answer:
Thank you, we have changed it.
- Line 138 - "and" should not be in italics.
Answer:
Corrected.
- 131
- Lines 146-157 - The authors need to justify the selection of the concentrations of both the bacterial suspensions and the nanoparticles solutions used in this study, especially since it is not expected that foods present such a high concentration of pathogenic bacteria, nor that just one bacterial species is present.
Answer:
Thank you for your comments. Yes the applied bacterium and NP concentrations are quite large, but some previous studies also used 106 CFU/ml. We are sure that contamination on the meat surface is mostly polymicrobial, but this aspect was outside of the scope of this study. Indeed this point will be important in future planned experiments, where the candidate TiO2 NPs will be fixed on a solid surface. In this case, the shielding of the pathogen either by other microbes, or with any food ingredients will be an important issue with practical relevance.
This is now justified in the Results and Discussion part:
- 219-221.
- Besides performing replicates (line 171), were the assays repeated in independent says?
Answer:
The three parallel experiments were carried out on the same day, but experiments were also repeated on a separate day. Results were the summary of these experiments. We made this clear now in the text.
- 166
- Line 178 - Although the nanoparticles have been characterized recently by the same research team, no cytotoxicity studies were performed. The authors have addressed this issue in lines 228-234, but the fact that they have chosen non-metals is not a guarantee of safety. This must be mentioned in the paper.
Answer:
Thank you for your reasonable comment. In this study we focused on the antibacterial feature of the tested NPs and we did not focus of their possible toxicity on the eukaryotic cells. In order to leave this question open till the results will be available we deleted the relevant sentence.
- 235 (deleted)
- The results and discussion section must be improved - the section is mainly used for results presentation, and could be further discussed considering other active surface components (considering authors statement in conclusions, lines 321-324) and not just only other TiO2 NPs.
Answer:
Thank you for your comment. In the results and discussion part we have focused on the comparison of the antibacterial efficacies of TiO2 NPs used by us and other authors. The available literature of this special segment is fairly narrow, but based on your opinion we have added some background information in that the efficacy of TiO2 NP to other NPs were also considered. Comparison of the different few results is more problematic as different studies have used different experiemntal setups, like liquid, agardiffusin and solid surfaces. Even though we tried to add some additional results.
The new informations are highlighted similarly to the newly added references in the reference list.

Reviewer 2 Report
This article concerns the antimicrobial efficacy of P and F co-doped TiO2 nanoparticles against seven foodborne pathogens. The study contains novelty regarding the synthesis of NPs, the number and species of pathogens involved and the findings about the susceptibility of the above bacteria after UVA exposure and during the photocatalytic reaction. In general, this is a well written and scientifically sound article from an experienced group of researchers based on proper methodology and with conclusions supported by the results which is of importance in scientific community as well as to the food industry.
To my opinion, the lack of statistical analysis is the major concern in order to further support the findings. Therefore, I’m recommending the inclusion of a) a section of statistical analysis in M&M and b) a comparison of the results between the various bacteria and also within each species (significance can be indicated in the figures).
Additionally,
Line 88. Number 2 should be in subscript in TiO2 formula
Line 158. How mixing was performed?
Line 200. Please add some more details regarding the last sentence.
Line 299-301. Please rephrase.
Regards
Author Response
Response to the comments and suggestions of Reviewer #2
Manuscript Number: Foods-1258146
Title:
“Antimicrobial efficacy and spectrum of Phosphorous-Fluorine co-doped TiO2 nanoparticles on the foodborne pathogenic bacteria Campylobacter jejuni, Salmonella Typhimurium, Enterohaemorrhagic E. coli, Yersinia enterocolitica, Shewanella putrefaciens, Listeria monocytogenes and Staphylococcus aureus”
First of all, thank you for the positive feedback concerning our manuscript submitted to Foods.
You can find our responses to your comments and suggestions below, while changes are highlighted in the text (turquoise):
General opinion:
This article concerns the antimicrobial efficacy of P and F co-doped TiO2 nanoparticles against seven foodborne pathogens. The study contains novelty regarding the synthesis of NPs, the number and species of pathogens involved and the findings about the susceptibility of the above bacteria after UVA exposure and during the photocatalytic reaction. In general, this is a well written and scientifically sound article from an experienced group of researchers based on proper methodology and with conclusions supported by the results which is of importance in scientific community as well as to the food industry.
Answer:
Authors thank for your positive opinion.
Comment:
To my opinion, the lack of statistical analysis is the major concern in order to further support the findings. Therefore, I’m recommending the inclusion of a) a section of statistical analysis in M&M and b) a comparison of the results between the various bacteria and also within each species (significance can be indicated in the figures).
Answer:
Thank you for your reasonable comment. We added the statistical part and constructed a novel Figure2 where significance is labelled.
Additional comments:
- Line 88. Number 2 should be in subscript in TiO2 formula
Answer:
Corrected.
- 89
- Line 158. How mixing was performed?
Answer:
Mixing /suspending was performed with a 1 ml pipette, that is now specified in the text.
- 152
- Line 200. Please add some more details regarding the last sentence.
Answer
According to your comment we added some background information concerning to this.
- 196-203
- Line 299-301. Please rephrase.
Answer:
We modified the relevant part of the section.
- 310-312
